# CALIBRATION OF NEURAL NETWORKS USING SPLINES

**Kartik Gupta[1,2], Amir Rahimi[1], Thalaiyasingam Ajanthan[1], Thomas Mensink[3], Cristian Sminchisescu[3], Richard Hartley[1,3]**
[1]Australian National University, [2]Data61, CSIRO, [3]Google Research
{kartik.gupta,amir.rahimi,thalaiyasingam.ajanthan}@anu.edu.au
{mensink,sminchisescu,richardhartley}@google.com

## ABSTRACT

Calibrating neural networks is of utmost importance when employing them in safety-critical applications where the downstream decision making depends on the predicted probabilities. Measuring calibration error amounts to comparing two empirical distributions. In this work, we introduce a *binning-free calibration measure* inspired by the classical Kolmogorov-Smirnov (KS) statistical test in which the main idea is to compare the respective cumulative probability distributions. From this, by approximating the empirical cumulative distribution using a differentiable function via splines, we obtain a *recalibration function*, which maps the network outputs to actual (calibrated) class assignment probabilities. The spline-fitting is performed using a held-out calibration set and the obtained recalibration function is evaluated on an unseen test set. We tested our method against existing calibration approaches on various image classification datasets and our spline-based recalibration approach consistently outperforms existing methods on KS error as well as other commonly used calibration measures.

## 1 INTRODUCTION

Despite the success of modern neural networks they are shown to be poorly calibrated (Guo et al. (2017)), which has led to a growing interest in the calibration of neural networks over the past few years (Kull et al. (2019); Kumar et al. (2019; 2018); Müller et al. (2019)). Considering classification problems, a classifier is said to be *calibrated* if the probability values it associates with the class labels match the true probabilities of correct class assignments. For instance, if an image classifier outputs 0.2 probability for the "horse" label for 100 test images, then out of those 100 images approximately 20 images should be classified as horse. It is important to ensure calibration when using classifiers for safety-critical applications such as medical image analysis and autonomous driving where the downstream decision making depends on the predicted probabilities.

One of the important aspects of machine learning research is the measure used to evaluate the performance of a model and in the context of calibration, this amounts to measuring the difference between two empirical probability distributions. To this end, the popular metric, Expected Calibration Error (ECE) (Naeini et al. (2015)), approximates the classwise probability distributions using histograms and takes an expected difference. This histogram approximation has a weakness that the resulting calibration error depends on the binning scheme (number of bins and bin divisions). Even though the drawbacks of ECE have been pointed out and some improvements have been proposed (Kumar et al. (2019); Nixon et al. (2019)), the histogram approximation has not been eliminated.[1]

In this paper, we first introduce a simple, *binning-free calibration measure* inspired by the classical Kolmogorov-Smirnov (KS) statistical test (Kolmogorov (1933); Smirnov (1939)), which also provides an effective visualization of the degree of miscalibration similar to the reliability diagram (Niculescu-Mizil & Caruana (2005)). To this end, the main idea of the KS-test is to compare the respective classwise cumulative (empirical) distributions. Furthermore, by approximating the empirical cumulative distribution using a differentiable function via splines (McKinley & Levine (1998)), we

---

[1]We consider metrics that measure classwise (top-$r$) calibration error (Kull et al. (2019)). Refer to section 2 for details.

obtain an *analytical recalibration function*[2] which maps the given network outputs to the actual class assignment probabilities. Such a direct mapping was previously unavailable and the problem has been approached indirectly via learning, for example, by optimizing the (modified) cross-entropy loss (Guo et al. (2017); Mukhoti et al. (2020); Müller et al. (2019)). Similar to the existing methods (Guo et al. (2017); Kull et al. (2019)) the spline-fitting is performed using a held-out calibration set and the obtained recalibration function is evaluated on an unseen test set.

We evaluated our method against existing calibration approaches on various image classification datasets and our spline-based recalibration approach consistently outperforms existing methods on KS error, ECE as well as other commonly used calibration measures. Our approach to calibration does not update the model parameters, which allows it to be applied on any trained network and it retains the original classification accuracy in all the tested cases.

## 2 NOTATION AND PRELIMINARIES

We abstract the network as a function $f_\theta : \mathcal{D} \to [0, 1]^K$, where $\mathcal{D} \subset \mathbb{R}^d$, and write $f_\theta(\mathbf{x}) = \mathbf{z}$. Here, $\mathbf{x}$ may be an image, or other input datum, and $\mathbf{z}$ is a vector, sometimes known as the vector of *logits*. In this paper, the parameters $\theta$ will not be considered, and we write simply $f$ to represent the network function. We often refer to this function as a *classifier*, and in theory this could be of some other type than a neural network.

In a classification problem, $K$ is the number of classes to be distinguished, and we call the value $z_k$ (the $k$-th component of vector $\mathbf{z}$) the *score* for the class $k$. If the final layer of a network is a *softmax* layer, then the values $z_k$ satisfy $\sum_{k=1}^{K} z_k = 1$, and $z_k \geq 0$. Hence, the $z_k$ are pseudo-probabilities, though they do not necessarily have anything to do with real probabilities of correct class assignments. Typically, the value $y^* = \arg\max_k z_k$ is taken as the (*top*-1) prediction of the network, and the corresponding score, $\max_k z_k$ is called the *confidence* of the prediction. However, the term confidence does not have any mathematical meaning in this context and we deprecate its use.

We assume we are given a set of training data $(\mathbf{x}_i, y_i)_{i=1}^n$, where $\mathbf{x}_i \in \mathcal{D}$ is an input data element, which for simplicity we call an image, and $y_i \in \mathcal{K} = \{1, \ldots, K\}$ is the so-called ground-truth label. Our method also uses two other sets of data, called *calibration data* and *test data*.

It would be desirable if the numbers $z_k$ output by a network represented true probabilities. For this to make sense, we posit the existence of joint random variables $(X, Y)$, where $X$ takes values in a domain $\mathcal{D} \subset \mathbb{R}^d$, and $Y$ takes values in $\mathcal{K}$. Further, let $Z = f(X)$, another random variable, and $Z_k = f_k(X)$ be its $k$-th component. Note that in this formulation $X$ and $Y$ are joint random variables, and the probability $P(Y \mid X)$ is *not* assumed to be 1 for single class, and 0 for the others.

A network is said to be *calibrated* if for every class $k$,

$$P(Y = k \mid Z = \mathbf{z}) = z_k . \tag{1}$$

This can be written briefly as $P(k \mid f(\mathbf{x})) = f_k(\mathbf{x}) = z_k$. Thus, if the network takes input $\mathbf{x}$ and outputs $\mathbf{z} = f(\mathbf{x})$, then $z_k$ represents the probability (given $f(\mathbf{x})$) that image $\mathbf{x}$ belongs to class $k$.

The probability $P(k \mid \mathbf{z})$ is difficult to evaluate, even empirically, and most metrics (such as ECE) use or measure a different notion called *classwise calibration* (Kull et al. (2019); Zadrozny & Elkan (2002)), defined as,

$$P(Y = k \mid Z_k = z_k) = z_k . \tag{2}$$

This paper uses this definition (2) of calibration in the proposed KS metric.

Calibration and accuracy of a network are different concepts. For instance, one may consider a classifier that simply outputs the class probabilities for the data, ignoring the input $\mathbf{x}$. Thus, if $f_k(\mathbf{x}) = z_k = P(Y = k)$, this classifier $f$ is calibrated but the accuracy is no better than the random predictor. Therefore, in calibration of a classifier, it is important that this is not done while sacrificing classification (for instance top-1) accuracy.

---

[2]Open-source implementation available at `https://github.com/kartikgupta-at-anu/spline-calibration`

**The top-$r$ prediction.**   The classifier $f$ being calibrated means that $f_k(\mathbf{x})$ is calibrated for each class $k$, not only for the top class. This means that scores $z_k$ for all classes $k$ give a meaningful estimate of the probability of the sample belonging to class $k$. This is particularly important in medical diagnosis where one may wish to have a reliable estimate of the probability of certain unlikely diagnoses.

Frequently, however, one is most interested in the probability of the top scoring class, the top-1 prediction, or in general the top-$r$ prediction. Suppose a classifier $f$ is given with values in $[0,1]^K$ and let $y$ be the ground truth label. Let us use $f^{(-r)}$ to denote the $r$-th top score (so $f^{(-1)}$ would denote the top score; the notation follows python semantics in which $A[-1]$ represents the last element in array $A$). Similarly we define $\max^{(-r)}$ for the $r$-th largest value. Let $f^{(-r)} : \mathcal{D} \to [0,1]$ be defined as

$$f^{(-r)}(\mathbf{x}) = \max_k^{(-r)} f_k(\mathbf{x}) , \quad \text{and} \quad y^{(-r)} = \begin{cases} 1 & \text{if } y = \arg\max_k^{(-r)} f_k(\mathbf{x}) \\ 0 & \text{otherwise .} \end{cases} \tag{3}$$

In words, $y^{(-r)}$ is 1 if the $r$-th top predicted class is the correct (ground-truth) choice. The network is calibrated for the top-$r$ predictor if for all scores $\sigma$,

$$P(y^{(-r)} = 1 \mid f^{(-r)}(\mathbf{x}) = \sigma) = \sigma . \tag{4}$$

In words, the conditional probability that the top-$r$-th choice of the network is the correct choice, is equal to the $r$-th top score.

Similarly, one may consider probabilities that a datum belongs to one of the top-$r$ scoring classes. The classifier is calibrated for being within-the-top-$r$ classes if

$$P\big( \textstyle\sum_{s=1}^r y^{(-s)} = 1 \,\big|\, \sum_{s=1}^r f^{(-s)}(\mathbf{x}) = \sigma \big) = \sigma . \tag{5}$$

Here, the sum on the left is 1 if the ground-truth label is among the top $r$ choices, 0 otherwise, and the sum on the right is the sum of the top $r$ scores.

## 3   KOLMOGOROV-SMIRNOV CALIBRATION ERROR

We now consider a way to measure if a classifier is classwise calibrated, including top-$r$ and within-top-$r$ calibration. This test is closely related to the Kolmogorov-Smirnov test (Kolmogorov (1933); Smirnov (1939)) for the equality of two probability distributions. This may be applied when the probability distributions are represented by samples.

We start with the definition of classwise calibration:

$$P(Y = k \mid f_k(X) = z_k) = z_k . \tag{6}$$
$$P(Y = k, \ f_k(X) = z_k) = z_k \, P(f_k(X) = z_k) , \quad \text{Bayes' rule .}$$

This may be written more simply but with a less precise notation as

$$P(z_k, \ k) = z_k \, P(z_k) .$$

**Motivation of the KS test.**   One is motivated to test the equality (or difference between) two distributions, defined on the interval $[0,1]$. However, instead of having a functional form of these distributions, one has only samples from them. Given samples $(\mathbf{x}_i, y_i)$, it is not straight-forward to estimate $P(z_k)$ or $P(z_k \mid k)$, since a given value $z_k$ is likely to occur only once, or not at all, since the sample set is finite. One possibility is to use histograms of these distributions. However, this requires selection of the bin size and the division between bins, and the result depends on these parameters. For this reason, we believe this is an inadequate solution.

The approach suggested by the Kolmogorov-Smirnov test is to compare the cumulative distributions. Thus, with $k$ given, one tests the equality

$$\int_0^\sigma P(z_k, k) \, dz_k = \int_0^\sigma z_k \, P(z_k) \, dz_k . \tag{7}$$

Writing $\phi_1(\sigma)$ and $\phi_2(\sigma)$ to be the two sides of this equation, the KS-distance between these two distributions is defined as $\mathrm{KS} = \max_\sigma |\phi_1(\sigma) - \phi_2(\sigma)|$. The fact that simply the maximum is used

here may suggest a lack of robustness, but this is a maximum difference between two integrals, so it reflects an accumulated difference between the two distributions.

To provide more insights into the KS-distance, let us a consider a case where $z_k$ consistently over or under-estimates $P(k \mid z_k)$ (which is usually the case, at least for top-1 classification (Guo et al. (2017))), then $P(k \mid z_k) - z_k$ has constant sign for all values of $z_k$. It follows that $P(z_k, k) - z_k P(z_k)$ has constant sign and so the maximum value in the KS-distance is achieved when $\sigma = 1$. In this case,

$$\text{KS} = \int_0^1 \left| P(z_k, k) - z_k P(z_k) \right| dz_k = \int_0^1 \left| P(k \mid z_k) - z_k \right| P(z_k) \, dz_k \ , \tag{8}$$

which is the expected difference between $z_k$ and $P(k \mid z_k)$. This can be equivalently referred to as the *expected calibration error* for the class $k$.

**Sampled distributions.**   Given samples $(\mathbf{x}_i, y_i)_{i=1}^N$, and a fixed $k$, one can estimate these cumulative distributions by

$$\int_0^\sigma P(z_k, k) \, dz_k \approx \frac{1}{N} \sum_{i=1}^N \mathbf{1}(f_k(\mathbf{x}_i) \leq \sigma) \times \mathbf{1}(y_i = k) \ , \tag{9}$$

where $\mathbf{1} : \mathcal{B} \to \{0, 1\}$ is the function that returns $1$ if the Boolean expression is true and otherwise $0$. Thus, the sum is simply a count of the number of samples for which $y_i = k$ and $f_k(\mathbf{x}_i) \leq \sigma$, and so the integral represents the proportion of the data satisfying this condition. Similarly,

$$\int_0^\sigma z_k \, P(z_k) \, dz_k \approx \frac{1}{N} \sum_{i=1}^N \mathbf{1}(f_k(\mathbf{x}_i) \leq \sigma) f_k(\mathbf{x}_i) \ . \tag{10}$$

These sums can be computed quickly by sorting the data according to the values $f_k(\mathbf{x}_i)$, then defining two sequences as follows.

$$\begin{aligned}
\tilde{h}_0 &= h_0 = 0 \ , \\
h_i &= h_{i-1} + \mathbf{1}(y_i = k)/N \ , \\
\tilde{h}_i &= \tilde{h}_{i-1} + f_k(\mathbf{x}_i)/N \ .
\end{aligned} \tag{11}$$

The two sequences should be the same, and the metric

$$\text{KS}(f_k) = \max_i \left| h_i - \tilde{h}_i \right| \ , \tag{12}$$

gives a numerical estimate of the similarity, and hence a measure of the degree of calibration of $f_k$. This is essentially a version of the Kolmogorov-Smirnov test for equality of two distributions.

**Remark.**   All this discussion holds also when $k < 0$, for top-$r$ and within-top-$r$ predictions as discussed in section 2. In (11), for instance, $f_{-1}(\mathbf{x}_i)$ means the top score, $f_{-1}(\mathbf{x}_i) = \max_k(f_k(\mathbf{x}_i))$, or more generally, $f_{-r}(\mathbf{x}_i)$ means the $r$-th top score. Similarly, the expression $y_i = -r$ means that $y_i$ is the class that has the $r$-th top score. Note when calibrating the top-1 score, our method is applied after identifying the top-1 score, hence, it does not alter the classification accuracy.

## 4   RECALIBRATION USING SPLINES

The function $h_i$ defined in (11) computes an empirical approximation

$$h_i \approx P(Y = k, f_k(X) \leq f_k(\mathbf{x}_i)) \ . \tag{13}$$

For convenience, the value of $f_k$ will be referred to as the *score*. We now define a continuous function $h(t)$ for $t \in [0, 1]$ by

$$h(t) = P(Y = k, f_k(X) \leq s(t)) \ , \tag{14}$$

where $s(t)$ is the $t$-th fractile score, namely the value that a proportion $t$ of the scores $f_k(X)$ lie below. For instance $s(0.5)$ is the median score. So, $h_i$ is an empirical approximation to $h(t)$ where $t = i/N$. We now provide the basic observation that allows us to compute probabilities given the scores.

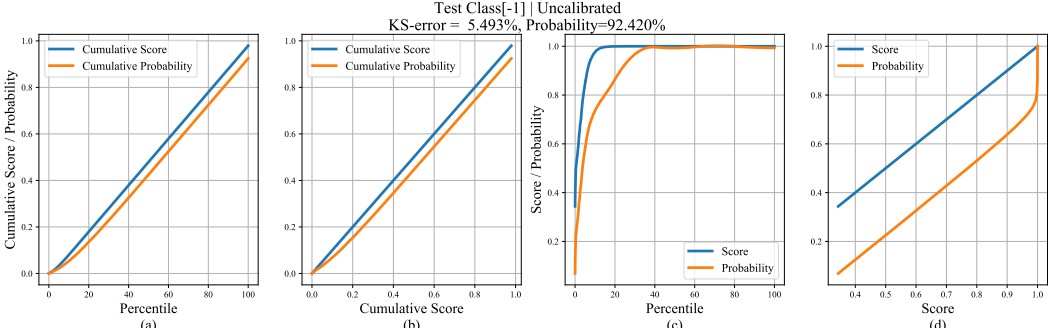

Figure 1: *Calibration graphs for an uncalibrated DenseNet-40 (Huang et al. (2017)) trained on CIFAR-10 for top-1 class with a KS error of 5.5%, and top-1 accuracy of 92.4% on the test set. Here (**a**) shows the plot of cumulative score and probability versus the fractile of the test set, (**b**) shows the same information with the horizontal axis warped so that the cumulative-score graph is a straight line. This is created as scatter plots of cumulative (score, score): blue and (score, probability): orange. If the network is perfectly calibrated, the probability line will be a straight line coincident with the (score, score) line. This shows that the network is substantially overestimating (score) the probability of the computation. (**c**) and (**d**) show plots of (non-cumulative) score and probability plotted against fractile, or score. How these plots are produced is described in section 4.*

**Proposition 4.1.** *If $h(t) = P(Y = k, f_k(X) \leq s(t))$ as in (14) where $s(t)$ is the $t$-th fractile score, then $h'(t) = P(Y = k \mid f_k(X) = s(t))$, where $h'(t) = dh/dt$.*

*Proof.* The proof relies on the equality $P(f_k(X) \leq s(t)) = t$. In words, since $s(t)$ is the value that a fraction $t$ of the scores are less than or equal, the probability that a score is less than or equal to $s(t)$, is (obviously) equal to $t$. See the supplementary material for a detailed proof.

Notice $h'(t)$ allows direct conversion from score to probability. Therefore, our idea is to approximate $h_i$ using a differentiable function and take the derivative which would be our recalibration function.

### 4.1 SPLINE FITTING

The function $h_i$ (shown in fig 1a) is obtained through sampling only. Nevertheless, the sampled graph is smooth and increasing. There are various ways to fit a smooth curve to it, so as to take derivatives. We choose to fit the sampled points $h_i$ to a cubic spline and take its derivative.

Given sample points $(u_i, v_i)_{i=1}^N$ in $\mathbb{R} \times \mathbb{R}$, easily available references show how to fit a smooth spline curve that passes directly through the points $(u_i, v_i)$. A very clear description is given in McKinley & Levine (1998), for the case where the points $u_i$ are equally spaced. We wish, however, to fit a spline curve with a small number of knot points to do a least-squares fit to the points. For convenience, this is briefly described here.

A cubic spline $v(u)$ is defined by its values at certain knot points $(\hat{u}_k, \hat{v}_k)_{k=1}^K$. In fact, the value of the curve at any point $u$ can be written as a linear function $v(u) = \sum_{k=1}^K a_k(u)\hat{v}_k = \mathbf{a}^\top(u)\,\hat{\mathbf{v}}$, where the coefficients $a_k$ depend on $u$.[3] Therefore, given a set of further points $(u_i, v_i)_{i=1}^N$, which may be different from the knot points, and typically more in number, least-squares spline fitting of the points $(u_i, v_i)$ can be written as a least-squares problem $\min_{\hat{\mathbf{v}}} \|\mathtt{A}(\mathbf{u})\hat{\mathbf{v}} - \mathbf{v}\|^2$, which is solved by standard linear least-squares techniques. Here, the matrix $\mathtt{A}$ has dimension $N \times K$ with $N > K$. Once $\hat{\mathbf{v}}$ is found, the value of the spline at any further points $u$ is equal to $v(u) = \mathbf{a}(u)^\top \hat{\mathbf{v}}$, a linear combination of the knot-point values $\hat{v}_k$.

Since the function is piecewise cubic, with continuous second derivatives, the first derivative of the spline is computed analytically. Furthermore, the derivative $v'(u)$ can also be written as a linear combination $v'(u) = \mathbf{a}'(u)^\top \hat{\mathbf{v}}$, where the coefficients $\mathbf{a}'(u)$ can be written explicitly.

Our goal is to fit a spline to a set of data points $(u_i, v_i) = (i/N, h_i)$ defined in (11), in other words, the values $h_i$ plotted against fractile score. Then according to Proposition 4.1, the derivative of the spline is equal to $P(k \mid f_k(X) = s(t))$. This allows a direct computation of the conditional probability that the sample belongs to class $k$.

---

[3]Here and elsewhere, notation such as $\mathbf{v}$ and $\mathbf{a}$ denotes the vector of values $v_i$ or $a_k$, as appropriate.

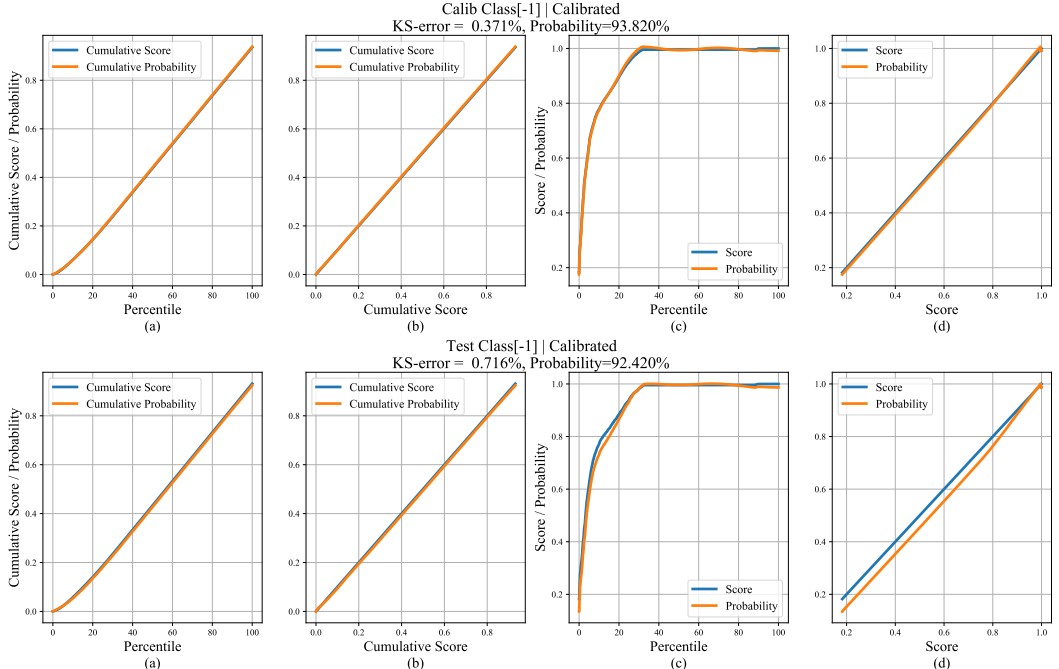

Figure 2: *The result of the spline calibration method, on the example given in fig 1 for top-1 calibration. A recalibration function $\gamma : \mathbb{R} \to \mathbb{R}$ is used to adjust the scores, replacing $f_k(\mathbf{x})$ with $\gamma(f_k(\mathbf{x}))$ (see section 4.2). As is seen, the network is now almost perfectly calibrated when tested on the "calibration" set (**top row**) used to calibrate it. In **bottom row**, the recalibration function is tested on a further set "test". It is seen that the result is not perfect, but much better than the one in fig 1d. It is also notable that the improvement in calibration is achieved without any loss of accuracy.*

Since the derivative of $h_i$ is a probability, one might constrain the derivative to be in the range $[0, 1]$ while fitting splines. This can be easily incorporated because the derivative of the spline is a linear expression in $\hat{v}_i$. The spline fitting problem thereby becomes a linearly-constrained quadratic program (QP). However, although we tested this, in all the reported experiments, a simple least-squares solver is used without the constraints.

## 4.2 RECALIBRATION

We suppose that the classifier $f = f_\theta$ is fixed, through training on the training set. Typically, if the classifier is tested on the training set, it is very close to being calibrated. However, if a classifier $f$ is then tested on a different set of data, it may be substantially mis-calibrated. See fig 1.

Our method of calibration is to find a further mapping $\gamma : [0, 1] \to [0, 1]$, such that $\gamma \circ f_k$ is calibrated. This is easily obtained from the direct mapping from score $f_k(\mathbf{x})$ to $P(k \mid f_k(\mathbf{x}))$ (refer to fig 1d). In equations, $\gamma(\sigma) = h'(s^{-1}(\sigma))$. The function $h'$ is known analytically, from fitting a spline to $h(t)$ and taking its derivative. The function $s^{-1}$ is a mapping from the given score $\sigma$ to its fractile $s^{-1}(\sigma)$. Note that, a held out calibration set is used to fit the splines and the obtained recalibration function $\gamma$ is evaluated on an unseen test set.

To this end, given a sample $\mathbf{x}$ from the test set with $f_k(\mathbf{x}) = \sigma$, one can compute $h'(s^{-1}(\sigma))$ directly in one step by interpolating its value between the values of $h'(f_k(\mathbf{x}_i))$ and $h'(f_k(\mathbf{x}_{i+1}))$ where $\mathbf{x}_i$ and $\mathbf{x}_{i+1}$ are two samples from the calibration set, with closest scores on either side of $\sigma$. Assuming the samples in the calibration set are ordered, the samples $\mathbf{x}_i$ and $\mathbf{x}_{i+1}$ can be quickly located using binary search. Given a reasonable number of samples in the calibration set, (usually in the order of thousands), this can be very accurate. In our experiments, improvement in calibration is observed in the test set with no difference to the accuracy of the network (refer to fig 2d). In practice, spline fitting is much faster than one forward pass through the network and it is highly scalable compared to learning based calibration methods.

## 5 RELATED WORK

**Modern calibration methods.**    In recent years, neural networks are shown to overfit to the Negative Log-Likelihood (NLL) loss and in turn produce overconfident predictions which is cited as the main reason for miscalibration (Guo et al. (2017)). To this end, modern calibration methods can be broadly categorized into 1) methods that adapt the training procedure of the classifier, and 2) methods that learn a recalibration function post training. Among the former, the main idea is to increase the entropy of the classifier to avoid overconfident predictions, which is accomplished via modifying the training loss (Kumar et al. (2018); Mukhoti et al. (2020); Seo et al. (2019)), label smoothing (Müller et al. (2019); Pereyra et al. (2017)), and data augmentation techniques (Thulasidasan et al. (2019); Yun et al. (2019); Zhang et al. (2018)).

On the other hand, we are interested in calibrating an already trained classifier that eliminates the need for training from scratch. In this regard, a popular approach is Platt scaling (Platt et al. (1999)) which transforms the outputs of a binary classifier into probabilities by fitting a scaled logistic function on a held out calibration set. Similar approaches on binary classifiers include Isotonic Regression (Zadrozny & Elkan (2001)), histogram and Bayesian binning (Naeini et al. (2015); Zadrozny & Elkan (2001)), and Beta calibration (Kull et al. (2017)), which are later extended to the multiclass setting (Guo et al. (2017); Kull et al. (2019); Zadrozny & Elkan (2002)). Among these, the most popular method is temperature scaling (Guo et al. (2017)), which learns a single scalar on a held out set to calibrate the network predictions. Despite being simple and one of the early works, temperature scaling is the method to beat in calibrating modern networks. Our approach falls into this category, however, as opposed to minimizing a loss function, we obtain a recalibration function via spline-fitting, which directly maps the classifier outputs to the calibrated probabilities.

**Calibration measures.**    Expected Calibration Error (ECE) (Naeini et al. (2015)) is the most popular measure in the literature, however, it has a weakness that the resulting calibration error depends on the histogram binning scheme such as the bin endpoints and the number of bins. Even though, some improvements have been proposed (Nixon et al. (2019); Vaicenavicius et al. (2019)), the binning scheme has not been eliminated and it is recently shown that any binning scheme leads to underestimated calibration errors (Kumar et al. (2019); Widmann et al. (2019)). Note that, there are binning-free metrics exist such as Brier score (Brier (1950)), NLL, and kernel based metrics for the multiclass setting (Kumar et al. (2018); Widmann et al. (2019)). Nevertheless, the Brier score and NLL measure a combination of calibration error and classification error (not just the calibration which is the focus). Whereas kernel based metrics, besides being computationally expensive, measure the calibration of the predicted probability vector rather than the classwise calibration error (Kull et al. (2019)) (or top-$r$ prediction) which is typically the quantity of interest. To this end, we introduce a binning-free calibration measure based on the classical KS-test, which has the same benefits as ECE and provides effective visualizations similar to reliability diagrams. Furthermore, KS error can be shown to be a special case of kernel based measures (Gretton et al. (2012)).

## 6 EXPERIMENTS

**Experimental setup.**    We evaluate our proposed calibration method on four different image-classification datasets namely CIFAR-10/100 (Krizhevsky et al. (2009)), SVHN (Netzer et al. (2011)) and ImageNet (Deng et al. (2009)) using LeNet (LeCun et al. (1998)), ResNet (He et al. (2016)), ResNet with stochastic depth (Huang et al. (2017)), Wide ResNet (Zagoruyko & Komodakis (2016)) and DenseNet (Huang et al. (2017)) network architectures against state-of-the-art methods that calibrate post training. We use the pretrained network logits[4] for spline fitting where we choose validation set as the calibration set, similar to the standard practice. Our final results for calibration are then reported on the test set of all datasets. Since ImageNet does not comprise the validation set, test set is divided into two halves: calibration set and test set. We use the natural cubic spline fitting method (that is, cubic splines with linear run-out) with 6 knots for all our experiments. Further experimental details are provided in the supplementary. For baseline methods namely: Temperature scaling, Vector scaling, Matrix scaling with ODIR (Off-diagonal and Intercept Regularisation), and Dirichlet calibration, we use the implementation of Kull *et al.* (Kull et al. (2019)).

---

[4]Pre-trained network logits are obtained from `https://github.com/markus93/NN_calibration`.

| Dataset | Model | Uncalibrated | Temp. Scaling | Vector Scaling | MS-ODIR | Dir-ODIR | **Ours (Spline)** |
|---|---|---|---|---|---|---|---|
| CIFAR-10 | Resnet-110 | 4.750 | 0.916 | 0.996 | 0.977 | 1.060 | **0.643** |
| | Resnet-110-SD | 4.102 | 0.362 | 0.430 | 0.358 | 0.389 | **0.269** |
| | DenseNet-40 | 5.493 | 0.900 | 0.890 | 0.897 | 1.057 | **0.773** |
| | Wide Resnet-32 | 4.475 | 0.296 | **0.267** | 0.305 | 0.291 | 0.367 |
| | Lenet-5 | 5.038 | 0.799 | 0.839 | 0.646 | 0.854 | **0.348** |
| CIFAR-100 | Resnet-110 | 18.481 | 1.489 | 1.827 | 2.845 | 2.575 | **0.575** |
| | Resnet-110-SD | 15.832 | **0.748** | 1.303 | 3.572 | 1.645 | 1.028 |
| | DenseNet-40 | 21.156 | **0.304** | 0.483 | 2.350 | 0.618 | 0.454 |
| | Wide Resnet-32 | 18.784 | 1.130 | 1.642 | 2.524 | 1.788 | **0.930** |
| | Lenet-5 | 12.117 | 1.215 | 0.768 | 1.047 | 2.125 | **0.391** |
| ImageNet | Densenet-161 | 5.721 | 0.744 | 2.014 | 4.723 | 3.103 | **0.406** |
| | Resnet-152 | 6.544 | 0.791 | 1.985 | 5.805 | 3.528 | **0.441** |
| SVHN | Resnet-152-SD | 0.852 | **0.552** | 0.570 | 0.573 | 0.607 | 0.556 |

Table 1: *KS Error (in %) for top-1 prediction (with lowest in bold and second lowest underlined) on various image classification datasets and models with different calibration methods. Note, our method consistently reduces calibration error to $< \mathbf{1}$% in almost all experiments, outperforming state-of-the-art methods.*

| Dataset | Model | Uncalibrated | Temp. Scaling | Vector Scaling | MS-ODIR | Dir-ODIR | **Ours (Spline)** |
|---|---|---|---|---|---|---|---|
| CIFAR-10 | Resnet-110 | 3.011 | 0.947 | 0.948 | 0.598 | 0.953 | **0.347** |
| | Resnet-110-SD | 2.716 | 0.478 | 0.486 | 0.401 | 0.500 | **0.310** |
| | DenseNet-40 | 3.342 | **0.535** | 0.543 | 0.598 | 0.696 | 0.695 |
| | Wide Resnet-32 | 2.669 | 0.426 | 0.369 | 0.412 | 0.382 | **0.364** |
| | Lenet-5 | 1.708 | 0.367 | **0.279** | 0.409 | 0.426 | 0.837 |
| CIFAR-100 | Resnet-110 | 4.731 | 1.401 | 1.436 | 0.961 | 1.269 | **0.371** |
| | Resnet-110-SD | 3.923 | **0.315** | 0.481 | 0.772 | 0.506 | 0.595 |
| | DenseNet-40 | 5.803 | 0.305 | 0.653 | 0.219 | **0.135** | 0.903 |
| | Wide Resnet-32 | 5.349 | 0.790 | 1.095 | 0.646 | 0.845 | **0.372** |
| | Lenet-5 | 2.615 | 0.571 | 0.439 | **0.324** | 0.799 | 0.587 |
| ImageNet | Densenet-161 | 1.689 | 1.044 | 1.166 | 1.288 | 1.321 | **0.178** |
| | Resnet-152 | 1.793 | 1.151 | 1.264 | 1.660 | 1.430 | **0.580** |
| SVHN | Resnet-152-SD | 0.373 | 0.226 | **0.216** | 0.973 | 0.218 | 0.492 |

Table 2: *KS Error (in %) for top-2 prediction (with lowest in bold and second lowest underlined) on various image classification datasets and models with different calibration methods. Again, our method consistently reduces calibration error to $< 1$% (less then $0.7$%, except for one case), in all experiments, the only one of the methods to achieve this.*

**Results.** We provide comparisons of our method using proposed KS error for the top most prediction against state-of-the-art calibration methods namely temperature scaling (Guo et al. (2017)), vector scaling, MS-ODIR, and Dirichlet Calibration (Dir-ODIR) (Kull et al. (2019)) in Table 1. Our method reduces calibration error to $1$% in almost all experiments performed on different datasets without any loss in accuracy. It clearly reflects the efficacy of our method irrespective of the scale of the dataset as well as the depth of the network architecture. It consistently performs better than the recently introduced Dirichlet calibration and Matrix scaling with ODIR (Kull et al. (2019)) in all the experiments. Note this is consistent with the top-1 calibration results reported in Table 15 of (Kull et al. (2019)). The closest competitor to our method is temperature scaling, against which our method performs better in 9 out of 13 experiments. Note, in the cases where temperature scaling outperforms our method, the gap in KS error between the two methods is marginal ($< \mathbf{0.3}$%) and our method is the second best. We provide comparisons using other calibration metrics in the supplementary.

From the practical point of view, it is also important for a network to be calibrated for top second/third predictions and so on. We thus show comparisons for top-2 prediction KS error in Table 2. An observation similar to the one noted in Table 1 can be made for the top-2 predictions as well. Our method achieves $< \mathbf{1}$% calibration error in all the experiments. It consistently performs well especially for experiments performed on large scale ImageNet dataset where it sets new *state-of-the-art for calibration*. We would like to emphasize here, though for some cases Kull *et al.* (Kull et al. (2019)) and Vector Scaling perform better than our method in terms of top-2 KS calibration error, overall (considering both top-1 and top-2 predictions) our method performs better.

## 7 CONCLUSION

In this work, we have introduced a binning-free calibration metric based on the Kolmogorov-Smirnov test to measure classwise or (within)-top-$r$ calibration errors. Our KS error eliminates the shortcomings of the popular ECE measure and its variants while accurately measuring the expected calibration error and provides effective visualizations similar to reliability diagrams. Furthermore, we introduced a simple and effective calibration method based on spline-fitting which does not involve any learning and yet consistently yields the lowest calibration error in the majority of our experiments. We believe, the KS metric would be of wide-spread use to measure classwise calibration and our spline method would inspire learning-free approaches to neural network calibration. We intend to focus on calibration beyond classification problems as future work.

## 8 ACKNOWLEDGEMENTS

The work is supported by the Australian Research Council Centre of Excellence for Robotic Vision (project number CE140100016). We would also like to thank Google Research and Data61, CSIRO for their support.

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

# Appendices

Here, we first provide the proof of our main result, discuss more about top-$r$ calibration and spline-fitting, and then turn to additional experiments.

## A    PROOF OF PROPOSITION 4.1

We first restate our proposition below.

**Proposition A.2.** *If $h(t) = P(Y = k, f_k(X) \leq s(t))$ as in (14) of the main paper where $s(t)$ is the $t$-th fractile score. Then $h'(t) = P(Y = k \mid f_k(X) = s(t))$, where $h'(t) = dh/dt$.*

*Proof.* The proof is using the fundamental relationship between the Probability Distribution Function (PDF) and the Cumulative Distribution Function (CDF) and it is provided here for completeness. Taking derivatives, we see (writing $P(k)$ instead of $P(Y = k)$):

$$
\begin{aligned}
h'(t) &= P(k, f_k(X) = s(t)) \cdot s'(t) \\
&= P(k \mid f_k(X) = s(t)) \cdot P(f_k(X) = s(t)) \cdot s'(t) \\
&= P(k \mid f_k(X) = s(t)) \cdot \frac{d}{dt}\big(P(f_k(X) \leq s(t))\big) \\
&= P(k \mid f_k(X) = s(t)) \cdot \frac{d}{dt}(t) \\
&= P(k \mid f_k(X) = s(t)) \ .
\end{aligned}
\tag{15}
$$

The proof relies on the equality $P(f_k(X) \leq s(t)) = t$. In words: $s(t)$ is the value that a fraction $t$ of the scores are less than or equal. This equality then says: the probability that a score is less than or equal to the value that a fraction $t$ of the scores lie below, is (obviously) equal to $t$. $\qquad \square$

## B    MORE ON TOP-$r$ AND WITHIN-TOP-$r$ CALIBRATION

In the main paper, definitions of top-$r$ and within-top-$r$ calibration are given in equations (4) and (5). Here, a few more details are given of how to calibrate the classifier $f$ for top-$r$ and within-top-$r$ calibration.

The method of calibration using splines described in this paper consists of fitting a spline to the cumulative accuracy, defined as $h_i$ in equation (11) in the main paper. For top-$r$ classification, the method is much the same as for the classification for class $k$. Equation (11) is replaced by sorting the data according to the $r$-th top score, then defining

$$
\begin{aligned}
\tilde{h}_0 &= h_0 = 0 \ , \\
h_i &= h_{i-1} + \mathbf{1}(y^{(-r)} = 1)/N \ , \\
\tilde{h}_i &= \tilde{h}_{i-1} + f^{(-r)}(\mathbf{x}_i)/N \ ,
\end{aligned}
\tag{16}
$$

where $y^{(-r)}$ and $f^{(-r)}(\mathbf{x}_i)$ are defined in the main paper, equation (3). These sequences may then be used both as a metric for the correct top-$r$ calibration and for calibration using spline-fitting as described.

For within-top-$r$ calibration, one sorts the data according to the sum of the top $r$ scores, namely $\sum_{s=1}^{r} f^{(-s)}(\mathbf{x}_i)$, then computes

$$\tilde{h}_0 = h_0 = 0 \, ,$$

$$h_i = h_{i-1} + \mathbf{1}\left(\sum_{s=1}^{r} y^{(-s)} = 1\right)\Big/N \, ,$$

(17)

$$\tilde{h}_i = \tilde{h}_{i-1} + \sum_{s=1}^{r} f^{(-s)}(\mathbf{x}_i)/N \, ,$$

As before, this can be used as a metric, or as the starting point for within-top-$r$ calibration by our method. Examples of this type of calibration (graphs for uncalibrated networks in fig 7 and fig 9) is given in the graphs provided in fig 8 and fig 10 for within-top-2 predictions and within-top-3 predictions respectively.

It is notable that if a classifier is calibrated in the sense of equation (1) in the main paper (also called multi-class-calibrated), then it is also calibrated for top-$r$ and within-top-$r$ classification.

## C   LEAST SQUARE SPLINE FITTING

Least-square fitting using cubic splines is a known technique. However, details are given here for the convenience of the reader. Our primary reference is (McKinley & Levine (1998)), which we adapt to least-squares fitting. We consider the case where the knot-points are evenly spaced.

We change notation from that used in the main paper by denoting points by $(x, y)$ instead of $(u, v)$. Thus, given knot points $(\hat{x}_i, \hat{y}_i)_{k=1}^{K}$ one is required to fit some points $(x_i, y_i)_{i=1}^{N}$. Given a point $x$, the corresponding spline value is given by $y = \mathbf{a}(x)^{\top}\mathtt{M}\hat{\mathbf{y}}$, where $\hat{\mathbf{y}}$ is the vector of values $\hat{y}_i$. The form of the vector $\mathbf{a}(x)$ and the matrix $\mathtt{M}$ are given in the following.

The form of the matrix $\mathtt{M}$ is derived from equation (25) in McKinley & Levine (1998). Define the matrices

$$\mathtt{A} = \begin{bmatrix} 4 & 1 & & & & \\ 1 & 4 & 1 & & & \\ & 1 & 4 & 1 & & \\ & & & \ddots & & \\ & & & 1 & 4 & 1 \\ & & & & 1 & 4 \end{bmatrix} \; ; \quad \mathtt{B} = \frac{6}{h^2}\begin{bmatrix} 1 & -2 & 1 & & & \\ & 1 & -2 & 1 & & \\ & & & \ddots & & \\ & & & 1 & -2 & 1 \end{bmatrix} \, ,$$

where $h$ is the distance between the knot points. These matrices are of dimensions $K - 2 \times K - 2$ and $K - 2 \times K$ respectively. Finally, let $\mathtt{M}$ be the matrix

$$\mathtt{M} = \begin{bmatrix} \mathbf{0}_K^{\top} \\ \mathtt{A}^{-1}\mathtt{B} \\ \mathbf{0}_K^{\top} \\ \mathtt{I}_{K \times K} \end{bmatrix} \, .$$

Here, $\mathbf{0}_K$ is a vector of zeros of length $K$, and $\mathtt{I}_{K \times K}$ is the identity matrix. The matrix $\mathtt{M}$ has dimension $2K \times K$.

Next, let the point $x$ lie between the knots $j$ and $j + 1$ and let $u = x - \hat{x}_j$. Then define the vector $\mathbf{v} = \mathbf{a}(x)$ by values

$$v_j = -u^3/(6h) + u^2/2 - hu/3 \, ,$$
$$v_{j+1} = u^3/(6h) - hu/6 \, ,$$
$$v_{j+K} = -u/h + 1 \, ,$$
$$v_{j+1+K} = u/h \, ,$$

with other entries equal to 0.

| Dataset | Image Size | # class | Calibration set | Test set |
|---------|------------|---------|-----------------|----------|
| CIFAR-10 | $32 \times 32$ | 10 | 5000 | 10000 |
| CIFAR-100 | $32 \times 32$ | 100 | 5000 | 10000 |
| SVHN | $32 \times 32$ | 10 | 6000 | 26032 |
| ImageNet | $224 \times 224$ | 1000 | 25000 | 25000 |

Table 3: *Dataset splits used for all the calibration experiments. Note, "calibration" set is used for spline fitting in our method and calibration for the baseline methods and then different methods are evaluated on "test" set.*

| Dataset | Model | Uncalibrated | Temp. Scaling | Vector Scaling | MS-ODIR | Dir-ODIR | **Ours (Spline)** |
|---------|-------|--------------|---------------|----------------|---------|----------|-------------------|
| | Resnet-110 | 1.805 | **0.097** | 0.176 | 0.140 | 0.195 | 0.277 |
| | Resnet-110-SD | 1.423 | 0.111 | 0.089 | 0.082 | **0.073** | 0.104 |
| CIFAR-10 | DenseNet-40 | 2.256 | 0.435 | 0.409 | 0.395 | **0.348** | 0.571 |
| | Wide Resnet-32 | 1.812 | 0.145 | **0.105** | 0.124 | 0.139 | 0.537 |
| | Lenet-5 | 3.545 | 0.832 | 0.831 | **0.631** | 0.804 | 0.670 |
| | Resnet-110 | 14.270 | 0.885 | 0.649 | 1.425 | 1.190 | **0.503** |
| | Resnet-110-SD | 12.404 | 0.762 | 1.311 | 2.120 | 1.588 | **0.684** |
| CIFAR-100 | DenseNet-40 | 15.901 | 0.437 | **0.368** | 2.205 | 0.518 | 0.724 |
| | Wide Resnet-32 | 14.078 | **0.414** | 0.548 | 1.915 | 1.099 | 1.017 |
| | Lenet-5 | 14.713 | 0.787 | 1.249 | 0.643 | 2.682 | **0.518** |
| ImageNet | Densenet-161 | 4.266 | 1.051 | 0.868 | 3.372 | 2.536 | **0.408** |
| | Resnet-152 | 4.851 | 1.167 | 0.776 | 4.093 | 2.839 | **0.247** |
| SVHN | Resnet-152-SD | 0.485 | 0.388 | 0.410 | 0.407 | 0.388 | **0.158** |

Table 4: **Within-top-2 predictions.** *KS Error (in %) within-top-2 prediction (with lowest in bold and second lowest underlined) on various image classification datasets and models with different calibration methods. Note, for this experiment we use 14 knots for spline fitting.*

Then the value of the spline is given by

$$y = \mathbf{a}(x)^\top \mathtt{M} \hat{\mathbf{y}} \; ,$$

as required. This allows us to fit the spline (varying the values of $\hat{\mathbf{y}}$) to points $(x_i, y_i)$ by least-squares fit, as described in the main paper.

The above description is for so-called *natural* (linear-runout) splines. For quadratic-runout or cubic-runout splines the only difference is that the first and last rows of matrix $\mathtt{A}$ are changed – see McKinley & Levine (1998) for details.

As described in the main paper, it is also possible to add linear constraints to this least-squares problem, such as constraints on derivatives of the spline. This results in a linearly-constrained quadratic programming problem.

# D  ADDITIONAL EXPERIMENTS

We first provide the experimental setup for different datasets in Table 3. Note, the calibration set is used for spline fitting in our method and then final evaluation is based on an unseen test set.

We also provide comparisons of our method against baseline methods for within-top-2 predictions (equation 5 of the main paper) in Table 4 using KS error. Our method achieves comparable or better results for within-top-2 predictions. It should be noted that the scores for top-3 $(f^{(-3)}(\mathbf{x}))$ or even top-4, top-5, etc., are very close to zero for majority of the samples (due to overconfidence of top-1 predictions). Therefore the calibration error for top-$r$ with $r > 2$ predictions is very close to zero and comparing different methods with respect to it is of little value. Furthermore, for visual illustration, we provide calibration graphs of top-2 predictions in fig 3 and fig 4 for uncalibrated and calibrated network respectively. Similar graphs for top-3, within-top-2, and within-top-3 predictions are presented in figures 5 – 10.

We also provide classification accuracy comparisons for different post-hoc calibration methods against our method if we apply calibration for all top-$1, 2, 3, \ldots, K$ predictions for $K$-class classification problem in Table 5. We would like to point out that there is negligible change in accuracy between the calibrated networks (using our method) and the uncalibrated ones.

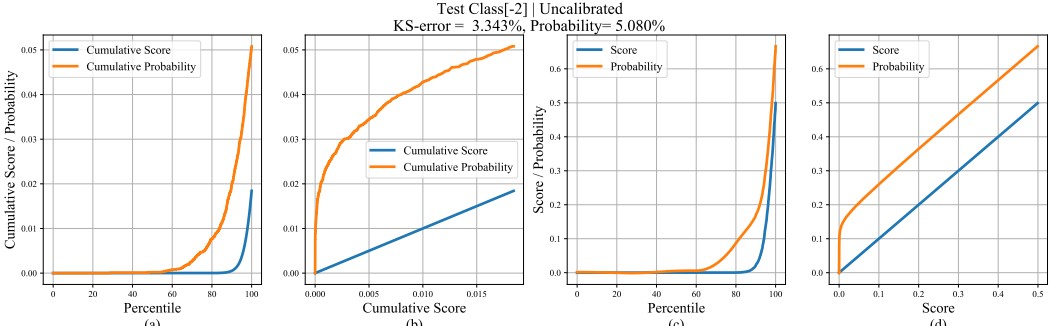

Figure 3: **Top-2 predictions, Uncalibrated.** *Calibration graphs for an uncalibrated DenseNet-40 (Huang et al. (2017)) trained on CIFAR-10 for top-2 class with a KS error of* 3.343% *on the test set. Here (**a**) shows the plot of cumulative score and probability versus the fractile of the test set, (**b**) shows the same information with the horizontal axis warped so that the cumulative-score graph is a straight line. This is created as scatter plots of cumulative (score, score): blue and (score, probability): orange. If the network is perfectly calibrated, the probability line will be a straight line coincident with the (score, score) line. This shows that the network is substantially overestimating (score) the probability of the computation. (**c**) and (**d**) show plots of (non-cumulative) score and probability plotted against fractile, or score. How these plots are produced is described in Section 4 of main paper.*

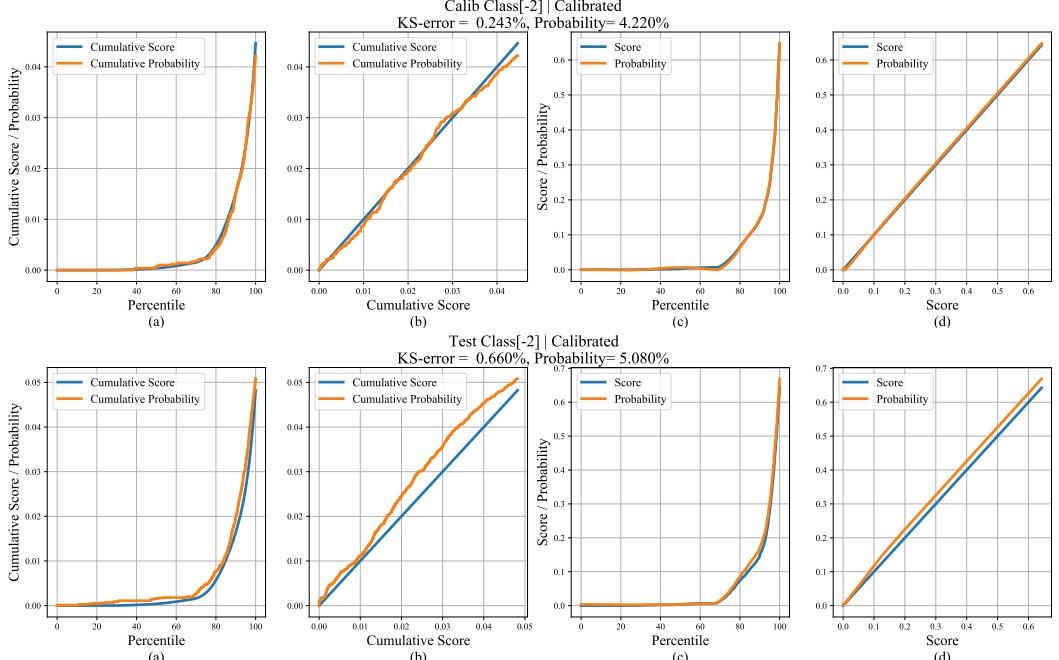

Figure 4: **Top-2 predictions, Calibrated.** *The result of the spline calibration method, on the example given in fig 3 for top-2 calibration. A recalibration function* $\gamma : \mathbb{R} \to \mathbb{R}$ *is used to adjust the scores, replacing* $f_k(\mathbf{x})$ *with* $\gamma(f_k(\mathbf{x}))$ *(see Section 4 of main paper). As is seen, the network is now almost perfectly calibrated when tested on the "calibration" set (**top row**) used to calibrate it. In **bottom row**, the recalibration function is tested on a further set "test". It is seen that the result is not perfect, but much better than the original results in fig 3d.*

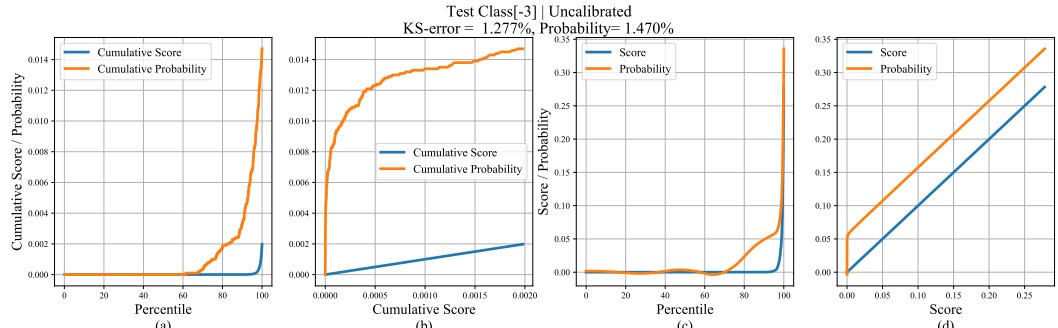

Figure 5: **Top-3 predictions, Uncalibrated.** *Calibration graphs for an uncalibrated DenseNet-40 trained on CIFAR-10 for top-3 class with a KS error of 1.277% on the test set. Here (**a**) shows the plot of cumulative score and probability versus the fractile of the test set, (**b**) shows the same information with the horizontal axis warped so that the cumulative-score graph is a straight line. (**c**) and (**d**) show plots of (non-cumulative) score and probability plotted against fractile, or score.*

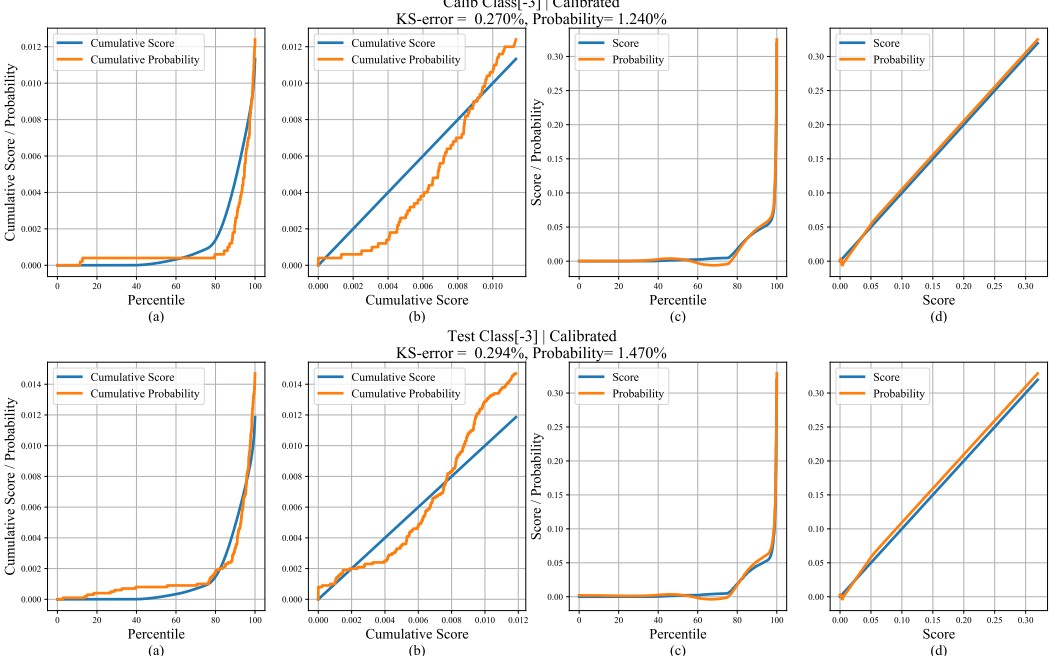

Figure 6: **Top-3 predictions, Calibrated.** *The result of the spline calibration method, on the example given in fig 5 for top-3 calibration. A recalibration function $\gamma : \mathbb{R} \to \mathbb{R}$ is used to adjust the scores, replacing $f_k(\mathbf{x})$ with $\gamma(f_k(\mathbf{x}))$. As is seen, the network is now almost perfectly calibrated when tested on the "calibration" set (**top row**) used to calibrate it. In **bottom row**, the recalibration function is tested on a further set "test". It is seen that the result is not perfect, but much better than the original results in fig 5d.*

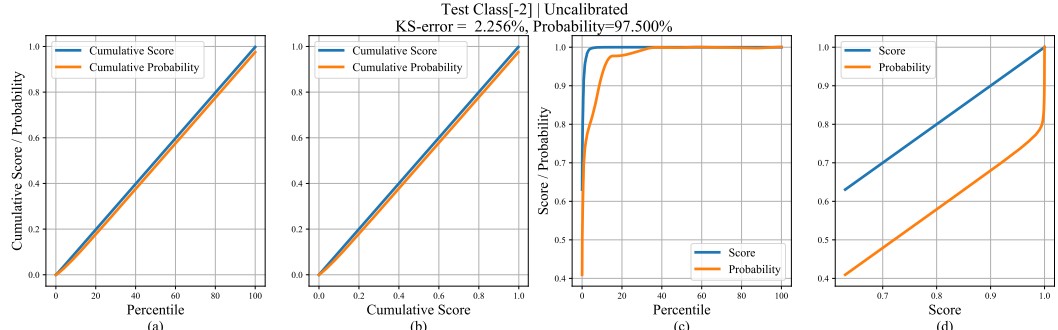

Figure 7: **Within-top-2 predictions, Uncalibrated.** *Calibration graphs for an uncalibrated DenseNet-40 trained on CIFAR-10 for within-top-2 predictions with a KS error of 2.256% on the test set. Here (**a**) shows the plot of cumulative score and probability versus the fractile of the test set, (**b**) shows the same information with the horizontal axis warped so that the cumulative-score graph is a straight line. (**c**) and (**d**) show plots of (non-cumulative) score and probability plotted against fractile, or score.*

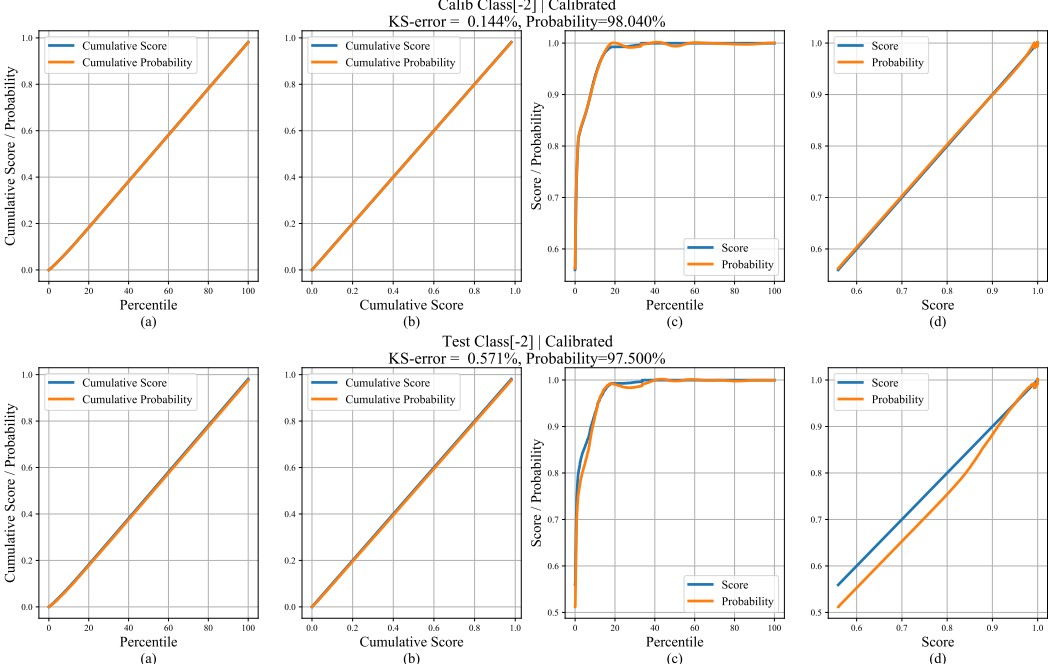

Figure 8: **Within-top-2 predictions, Calibrated.** *The result of the spline calibration method, on the example given in fig 7 for within-top-2 calibration. A recalibration function $\gamma : \mathbb{R} \to \mathbb{R}$ is used to adjust the scores, replacing $f_k(\mathbf{x})$ with $\gamma(f_k(\mathbf{x}))$. As is seen, the network is now almost perfectly calibrated when tested on the "calibration" set (**top row**) used to calibrate it. In **bottom row**, the recalibration function is tested on a further set "test". It is seen that the result is not perfect, but much better than the original results in fig 7d.*

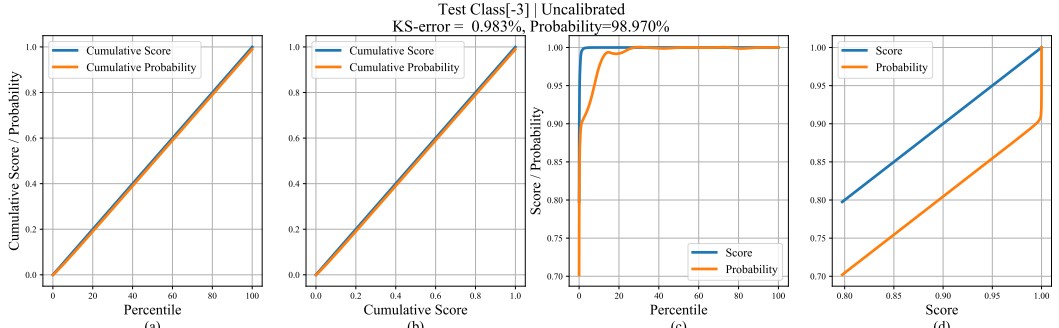

Figure 9: **Within-top-3 predictions, Uncalibrated.** *Calibration graphs for an uncalibrated DenseNet-40 trained on CIFAR-10 for within-top-3 predictions with a KS error of 0.983% on the test set. Here (**a**) shows the plot of cumulative score and probability versus the fractile of the test set, (**b**) shows the same information with the horizontal axis warped so that the cumulative-score graph is a straight line. (**c**) and (**d**) show plots of (non-cumulative) score and probability plotted against fractile, or score.*

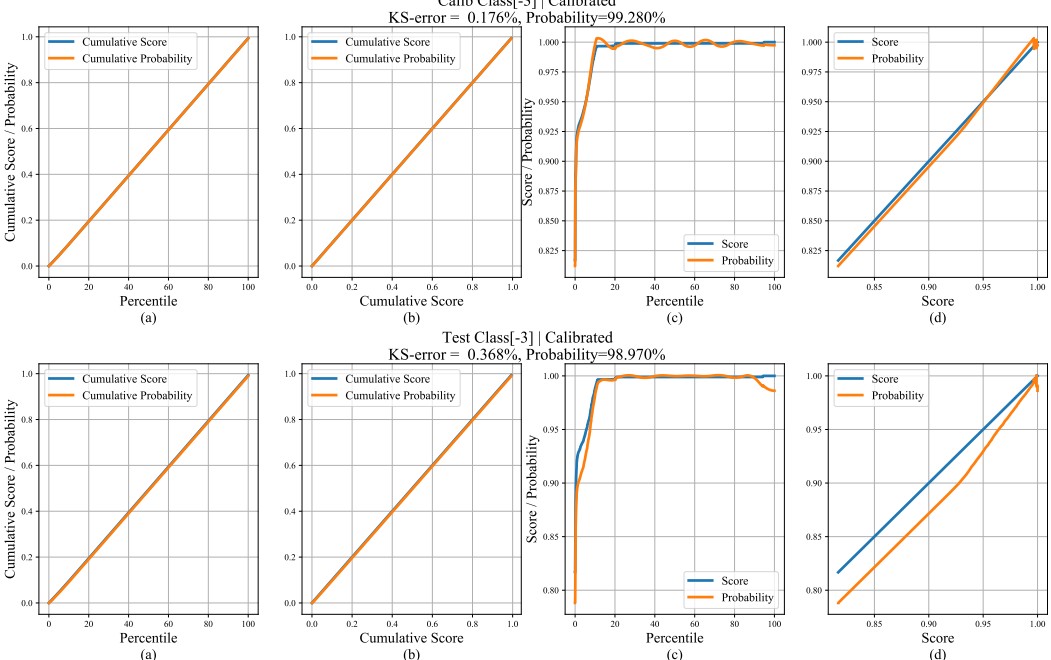

Figure 10: **Within-top-3 predictions, Calibrated.** *The result of the spline calibration method, on the example given in fig 9 for within-top-3 calibration. A recalibration function $\gamma : \mathbb{R} \to \mathbb{R}$ is used to adjust the scores, replacing $f_k(\mathbf{x})$ with $\gamma(f_k(\mathbf{x}))$. As is seen, the network is now almost perfectly calibrated when tested on the "calibration" set (**top row**) used to calibrate it. In **bottom row**, the recalibration function is tested on a further set "test". It is seen that the result is not perfect, but much better than the original results in fig 9d.*

| Dataset | Model | Uncalibrated | Temp. Scaling | Vector Scaling | MS-ODIR | Dir-ODIR | **Ours (Spline)** |
|---------|-------|--------------|---------------|----------------|---------|----------|-------------------|
| | Resnet-110 | **93.56** | **93.56** | 93.50 | 93.53 | 93.52 | 93.55 |
| | Resnet-110-SD | 94.04 | 94.04 | 94.04 | 94.18 | **94.20** | 94.05 |
| CIFAR-10 | DenseNet-40 | 92.42 | 92.42 | 92.50 | **92.52** | 92.47 | 92.31 |
| | Wide Resnet-32 | 93.93 | 93.93 | 94.21 | **94.22** | **94.22** | 93.76 |
| | Lenet-5 | 72.74 | 72.74 | 74.48 | 74.44 | **74.52** | 72.64 |
| | Resnet-110 | 71.48 | 71.48 | 71.58 | 71.55 | **71.62** | 71.50 |
| | Resnet-110-SD | 72.83 | 72.83 | **73.60** | 73.53 | 73.14 | 72.81 |
| CIFAR-100 | DenseNet-40 | 70.00 | 70.00 | 70.13 | **70.40** | 70.24 | 70.17 |
| | Wide Resnet-32 | 73.82 | 73.82 | 73.87 | **74.05** | 73.99 | 73.74 |
| | Lenet-5 | 33.59 | 33.59 | 36.42 | **37.58** | 37.52 | 33.55 |
| ImageNet | Densenet-161 | 77.05 | 77.05 | 76.72 | 77.15 | **77.19** | 77.05 |
| | Resnet-152 | 76.20 | 76.20 | 75.87 | 76.12 | **76.24** | 76.07 |
| SVHN | Resnet-152-SD | 98.15 | 98.15 | 98.13 | 98.12 | **98.19** | 98.17 |

Table 5: *Classification (top-1) accuracy (with highest in bold and second highest underlined) post calibration on various image classification datasets and models with different calibration methods. Note, only a negligible change in accuracy is observed in our method compared to the uncalibrated networks.*

| Dataset | Model | Uncalibrated | Temp. Scaling | Vector Scaling | MS-ODIR | Dir-ODIR | **Ours (Spline)** |
|---------|-------|--------------|---------------|----------------|---------|----------|-------------------|
| | Resnet-110 | 4.750 | 1.224 | 1.092 | 1.276 | 1.240 | **1.011** |
| | Resnet-110-SD | 4.135 | 0.777 | 0.752 | **0.684** | 0.859 | 0.992 |
| CIFAR-10 | DenseNet-40 | 5.507 | **1.006** | 1.207 | 1.250 | 1.268 | 1.389 |
| | Wide Resnet-32 | 4.512 | 0.905 | **0.852** | 0.941 | 0.965 | 1.003 |
| | Lenet-5 | 5.188 | 1.999 | 1.462 | 1.504 | **1.300** | 1.333 |
| | Resnet-110 | 18.480 | 2.428 | 2.722 | 3.011 | 2.806 | **1.868** |
| | Resnet-110-SD | 15.861 | **1.335** | 2.067 | 2.277 | 2.046 | 1.766 |
| CIFAR-100 | DenseNet-40 | 21.159 | **1.255** | 1.598 | 2.855 | 1.410 | 2.114 |
| | Wide Resnet-32 | 18.784 | **1.667** | 1.785 | 2.870 | 2.128 | 1.672 |
| | Lenet-5 | 12.117 | 1.535 | 1.350 | 1.696 | 2.159 | **1.029** |
| ImageNet | Densenet-161 | 5.720 | 2.059 | 2.637 | 4.337 | 3.989 | **0.798** |
| | Resnet-152 | 6.545 | 2.166 | 2.641 | 5.377 | 4.556 | **0.913** |
| SVHN | Resnet-152-SD | 0.877 | 0.675 | **0.630** | 0.646 | 0.651 | 0.832 |

Table 6: *ECE for top-1 predictions (in %) using 25 bins (with lowest in bold and second lowest underlined) on various image classification datasets and models with different calibration methods. Note, for this experiment we use 13 knots for spline fitting.*

For the sake of completeness, we present calibration results using the existing calibration metric, Expected Calibration Error (ECE) (Naeini et al. (2015)) in Table 6. We would like to reiterate the fact that ECE metric is highly dependent on the chosen number of bins and thus does not really reflect true calibration performance. To reflect the efficacy of our proposed calibration method, we also present calibration results using other calibration metrics such as recently proposed binning free measure KDE-ECE (Zhang et al. (2020)), MCE (Maximum Calibration Error) (Guo et al. (2017)) and Brier Scores for top-1 predictions on ImageNet dataset in Table 7. Since, the original formulation of Brier Score for multi-class predictions is highly biased on the accuracy and is approximately similar for all calibration methods, we hereby use top-1 Brier Score which is the mean squared error between top-1 scores and ground truths for the top-1 predictions (1 if the prediction is correct and 0 otherwise). It can be clearly observed that our approach consistently outperforms all the baselines on different calibration measures.

| Calibration Metric | Model | Uncalibrated | Temp. Scaling | MS-ODIR | Dir-ODIR | **Ours (Spline)** |
|---|---|---|---|---|---|---|
| **KDE-ECE** | Densenet-161 | 0.03786 | 0.01501 | 0.02874 | 0.02979 | **0.00637** |
| | Resnet-152 | 0.04650 | 0.01864 | 0.03448 | 0.03488 | **0.00847** |
| **MCE** | Densenet-161 | 0.13123 | **0.05442** | 0.09077 | 0.09653 | 0.06289 |
| | Resnet-152 | 0.15930 | 0.09051 | 0.11201 | 0.09868 | **0.04950** |
| **Brier Score** | Densenet-161 | 0.12172 | 0.11852 | 0.11982 | 0.11978 | **0.11734** |
| | Resnet-152 | 0.12626 | 0.12145 | 0.12406 | 0.12308 | **0.12034** |

Table 7: *Calibration Error using other different metrics such as binning-free KDE-ECE (Zhang et al. (2020)), MCE (Maximum Calibration Error) (Guo et al. (2017)) and Brier Score for top-1 predictions (with lowest in bold and and second lowest underlined) on ImageNet dataset with different calibration methods. Note, for this experiment we use 6 knots for spline fitting.*

