# OpenReview forum: "Calibration of Neural Networks using Splines"
_ICLR.cc/2021/Conference — ICLR 2021 Poster_

### Official Review · AnonReviewer1 · 2020-10-24
**Solid work on an important topic**

**Rating:** 7
**Confidence:** 4

**Review:**

The paper presents a post-hoc calibration method for deep neural net classification. The method proposes to first reduces the well-known ECE score to a special case of the Kolmogorov-Smirnov (KS) test, and this way solves the dependency of ECE on the limiting binning assumption. The method proposes next to recalibrate the classification probabilities by fitting a cubic spline to the KS test score.

Strengths:
 * The proposed approach is a clearly novel solution to an essential problem for the safety-critical use of deep learning.
 * The paper presents the material in a very clear and neat way, following an excellent scientific writing practice.
 * Both the KS test based treatment of network calibration and the elegant way to improve the calibration by cubic spline fitting are interesting and likely to attract the attention of the deep learning uncertainty community.
 * The paper presents an exhaustive set of experiments that report performances of quite deep state-of-the-art network architectures on four different benchmark tasks. It also provides comparisons against a decent number of state-of-the-art calibration methods. The results demonstrate the benefits of the proposed method.

(Minor) Weaknesses: The paper can be improved with a few small adjustments in the presentation:

 * The so-called "python semantics" notation in Sec 2 only introduces complexity and does not bring any concrete value to the story line. What is wrong with denoting the r-th top score simply as f^{(n)}. Why do we need the minus? We can simply assume that the first element of the sorted list is the top score.

 * Why do we need Proposition 4.1 and its proof (sketch and full) in this paper? It only echoes the standard relationship between a PDF and CDF. The former is the derivative of the latter.

 * Although the results in Tables 1 and 2 are groundbreaking in favor of the proposed method, they are calibration scores based on the KS score. That is only one way of measuring the calibratedness of a network, and furthermore that is the score the proposed method is maximizing. It would be interesting to see how well the proposed method generalizes across other calibration scores that may highlight complementary aspects of the uncertainty treatment capabilities of the predictor, such as the Brier score.

Overall, this is a solid piece of work that qualifies to appear at the ICLR'21 proceedings.

---

> ### Author Response · Authors · 2020-11-20
> **AR1: Thank you for the positive feedback.**
>
> We appreciate that the reviewer finds that our method is a novel solution and the paper follows an excellent scientific writing practice. Below we address the reviewer’s comments. If any answer requires further clarification to the reviewer, we will make our best effort to improve the clarity.
>
> ### Python semantics
> - We use the “minus” notation to distinguish top-r score with the r-th class score. So in our notation, f^{(-r)} denotes the r-th top score while f^{(r)} denotes the score of the r-th class.
>
> ### Proposition 4.1
> - We agree that Proposition 4.1 echoes the standard relationship between PDF and CDF. However, we think it might not be obvious to all the readers and we wanted to make it absolutely clear by having the proof.
>
> ### Other calibration measures
> - We note that ECE results are already reported in Table 6. While we share the same opinion as the reviewer, we want to clarify that Brier score and NLL are not good measures of the calibration error. The reason is that both Brier score and NLL can be zero only if the classification accuracy is 100%, however, it is not necessary for a classifier to have 100% accuracy for it to be perfectly calibrated. Nevertheless, we computed different calibration measures (including the Brier score) on ImageNet for different networks and the results are reported below. In summary, spline fitting outperforms other methods in most cases. We have also reported these results in our updated manuscript in Table 7.
>
> **KDE-ECE**[a]:
>
> |Network|Uncalibrated|Temp. Scaling|MS-ODIR|Dir-ODIR|Ours(Spline)|
> |-|-|-|-|-|-|
> |Densenet-161|0.03786|0.01501|0.02874|0.02979|**0.00637**|
> |Resnet-152|0.04650|0.01864|0.03448|0.03488|**0.00847**|
>
> **MCE (Maximum Calibration Error)**[b]:
>
> |Network|Uncalibrated|Temp. Scaling|MS-ODIR|Dir-ODIR|Ours(Spline)|
> |-|-|-|-|-|-|
> |Densenet-161|0.13123|**0.05442**|0.09077|0.09653|0.06289|
> |Resnet-152|0.15930|0.09051|0.11201|0.09868|**0.04950**|
>
> **BRIER Score**:
>
> |Network|Uncalibrated|Temp. Scaling|MS-ODIR|Dir-ODIR|Ours(Spline)|
> |-|-|-|-|-|-|
> |Densenet-161|0.00033|0.00032|0.00032|0.00032|0.00032|
> |Resnet-152|0.00034|0.00034|0.00033|0.00033|0.00033|
>
> - One can clearly notice that Brier score is almost same for all methods even for uncalibrated models, clearly illustrating it as not a good measure for calibration. This observation is infact not surprising since Brier Score in essence also measures how accurate a function is. Since, we specifically calibrate top-1 scores, we also evaluate top-1 Brier score which is mean squared error between top-1 scores and accuracy and the results are reported as follows:
>
> |Network|Uncalibrated|Temp. Scaling|MS-ODIR|Dir-ODIR|Ours(Spline)|
> |-|-|-|-|-|-|
> |Densenet-161|0.12172|0.11852|0.11982|0.11978|**0.11734**|
> |Resnet-152|0.12626|0.12145|0.12406|0.12308|**0.12034**|
>
> ### References
> [a] Jize Zhang, Bhavya Kailkhura, and T Han. "Mix-n-Match: Ensemble and compositional methods for uncertainty calibration in deep learning.", ICML 2020,
>
> [b] Chuan Guo, Geoff Pleiss, Yu Sun, and Kilian Q Weinberger. On calibration of modern neural networks. ICML, 2017.

---

### Official Review · AnonReviewer4 · 2020-10-27
**An ok submission but incremental.**

**Rating:** 5
**Confidence:** 4

**Review:**

The paper proposes a post re-calibration spline-based approach for the re-calibration of multiclass predictions.  Experiment results on image datasets are provided and evaluated according to the Kolmogorov–Smirnov (KS) statistic.

**Strengths**:
 - The proposed binning-free calibration measure is a good idea and widely used in binary classification problems

- As expected, experimental results demonstrate that optimizing calibration with a spline-based approach results in better calibration

**Weaknesses**:

*Inconsistent notation*:
-  The authors should stick to either $x $ or $X$
-  The definition of the KS statistic is inconsistent with standard formulations Eq (8). Also, the paper claims to obtain the maximum value at $\sigma=1$; this is a strong assumption.

*Weak experiments*:
- Why KS statistic and not Wasserstein or calibration slope?
-  Eq (11) assumes $f_k(x_i)$ are distinct, which is not always the case.
- Calibration and accuracy are orthogonal concerns, where the proposed post re-calibration approach is prone to overfit on calibration at the expense of accuracy (and this is the case, as experimental results show a loss in accuracy)
-  Demonstrating an approach that accounts for the calibration-accuracy tradeoff is crucial
- Extending the proposed approach to regression problems would strengthen the submission
- A  qualitative discussion on:
1)  Why Temp. Scaling is better calibrated in some instances
2)  Why calibration results of proposed solution (and alternatives) vary across different network architectures for the same dataset
3) Why the proposed approach drops in performance between top-1 and top-2 predictions
- The proposed re-calibration approach may not scale well with large datasets; a computational complexity comparison against alternatives is crucial

---

> ### Author Response · Authors · 2020-11-20
> **AR4: Our work is found to be “original and elegant” by other reviewers. {Response to AR4 [1/2]}**
>
> We thank the reviewer for the feedback and we appreciate that the reviewer finds that the introduced binning-free calibration measure is a good idea and our spline fitting method yields better calibration. We would like to emphasize that our method is found to be “elegant and technically correct with excellent scientific practice” by the other reviewers and AR2 noted that “the final method is far from obvious with several original contributions”. Below, we address the reviewer’s concerns and we have added the requested discussions in the updated version of the paper. If any answer requires further clarification to the reviewer, we will make our best effort to improve the clarity.
>
> ### The authors should stick to either $x$ or $X$
> - Following the standard notation, $X$ denotes the random variable and $x$ denotes an assignment to the random variable $X$.
>
> ### The definition of the KS statistic
> - The standard definition of the KS statistic is given after Eq. 7 and this is the definition that is used in our computation as noted in Eq. 12.
>
> ### Confusion with Eq. 8
> - We would like to clarify that _we do not make any assumptions_ and Eq. 8 is only used to provide insights into the KS metric. Specifically, KS metric becomes the same as the expected calibration error if $P(k|z_k) - z_k$ has the same sign for all samples. This case means the scores provided by the network consistently overestimate or underestimate the probability, which is usually the case (see eg., Guo et al). Even though this condition is satisfied in most cases, it is not necessary for our method or for the introduced binning-free calibration metric.
>
> ### Why KS statistic and not Wasserstein or calibration slope?
> - KS is chosen over Wasserstein distance, as we are interested in measuring calibration of each class score (or top-k score) individually, ie, *classwise calibration error*. Furthermore, as shown in our paper, KS provides an effective way to obtain a recalibration function using spline fitting. If one wishes to consider the full probability vector, Wasserstein distance may be preferred. Note that we also provided the ECE measure (Table 6) in addition to the KS statistic to show the generality of our method.
> - We’re not sure what is meant by “calibration slope”.
>
> ### Distinct values for $f_k(x_i)$
> - Eq. 11 does _not_ assume distinct values for scores $f_k(x_i)$ and if the scores are equal for two samples ties can be broken arbitrarily.
>
> ### Calibration vs accuracy
> - Since our spline based recalibration function is applied after identifying the top-1 score (or in general top-r), technically our method does not alter the classification accuracy. However, even if one wishes to evaluate the classification accuracy after applying the recalibration function, the change in accuracy with spline fitting is negligible (_<0.2%_) and it sometimes even improves the accuracy of the uncalibrated model as noted in Table 5.
>
> ### Extending to regression problems
> - We fully agree that calibration of neural networks is important beyond classification. We have highlighted this and included binning free regression calibration as a possible direction for future work. In this work, similar to the majority of the previous works on calibration, we focus on classification problems.
>
> ### Temperature scaling vs spline fitting
> - Since calibration is a statistical process, it is possible that in certain circumstances one method may work better than the other. To this end, we are not sure about the reason for temperature scaling being better in some cases. According to our experiments, our method performs better than temperature scaling in most cases.
>
> ### Calibration results vary across different networks
> - Different architectures have different calibration properties, for example, Resnet is known to be more prone to be overconfident than Lenet, see (Guo et al.) So, even for the same dataset, different architectures behave differently, have different accuracies and can have different calibration performance.
>
> ### Calibrating top-1 vs top-2 predictions
> - The distributions of top-1 and top-2 scores are very different, and we observed that the top-2 scores are very close to 0 in many cases. Since the values are very small, spline fitting may not seem to be as accurate as for the top-1 case. But it is important to note in Table 2 that our method still achieves (_<1%_) calibration error even for top-2 predictions and ranks as the best method in majority of the cases. One could potentially vary the spline fitting parameters (such as the number of knots) to obtain better results. Note, for majority of our experiments only 6 knots are used unless stated otherwise.

---

> > ### Author Response · Authors · 2020-11-20
> > **Other comments {Response to AR4[2/2]}**
> >
> > ### Scalability of spline fitting
> > - Both in theory as well as in practice, spline fitting is very fast (about 0.4 seconds on the ImageNet calibration set with 25000 images). Much faster than one forward pass through the network and it is learning free. See also our response to AR2. The most complex step of spline fitting is sorting which is very efficient. So we believe, our method would be highly scalable compared to methods that require learning.

---

### Official Review · AnonReviewer3 · 2020-10-28
**The paper presents an interesting contribution with potential applicability in the deep learning community.**

**Rating:** 8
**Confidence:** 4

**Review:**

The authors present a binning-free calibration measure from a Kolmogorov-Smirnov-based test. Besides, the cumulative probability distribution is estimated using a spline-based fitting from percentiles. The approach allows correcting the probability estimation from trained deep learning models. The paper is clear and well-founded.

---

> ### Author Response · Authors · 2020-11-20
> **AR3: Thank you for the positive feedback.**
>
> We thank the reviewer for the positive feedback and appreciate that the reviewer finds our paper to be clearly written and the proposed method to be well-founded.

---

### Official Review · AnonReviewer2 · 2020-10-29
**Elegant calibration method, beautifully described. Experiments may need more explaining.**

**Rating:** 8
**Confidence:** 4

**Review:**

Pros: originality, clarity, technical correctness.
Cons: experiments need some clarifications.

Calibration typically relies heavily on binning the data, both for the calibration itself  (Histogram Binning) and how to measure its quality (ECE). Thus both operations suffer from sampling issues that are a cause for both bias and variance. The idea to rather perform the analysis using cumulative distributions would seem obvious, as it has been tried on so many other problems, but I have not seen it used for calibration. They do it for both calibration and its measure:
-	The use of the Kolmogorov-Smirnov test to measure calibration between the target and the output distributions.
-	Spline fitting of the cumulative distribution to compute its derivative
As the implementation details are far from obvious , there are several original contributions, especially in implementing the splines.

I found the description both very clear and concise, switching between intuition and equations.
The only part I found confusing is the second paragraph of section 4.2 (“One method of calibration..”). Fortunately, the next paragraph gives a very simple intuitive explanation by just stating how it is implemented.

Experiments are very comprehensive and show improvements over Temp scaling and other methods. However, there are also some results that contradict previously reported experiments and need to be clarified:
-	Besides Temp scaling, the main other methods are borrowed from Kull et al, so one could expect some consistency. However ECE results from Table 6 look very different from Table 3 in Kull et al. I assume these are different types of ECE: confidence vs. class-wise? In Table 1, KS result for the ODIR methods of Kull et Al are much worse than Temp scaling, in particular for CIFAR-100 and Imagenet. This contradicts results reported in Table 2 (this paper) and Table 3 (Kull et al) and should be explained.

-	I am no expert in image classification,  but the 70% accuracy reported for CIFAR-100 seems way below current numbers, which have exceeded 80% since 2017, in particular for the proposed architectures (for instance Wide Resnet or DenseNet) https://benchmarks.ai/cifar-100.  However these numbers seem to be consistent with what is reported by Kull et al (Table 18 in https://arxiv.org/pdf/1910.12656.pdf), so I assume the issue comes from borrowing their architecture and scores. While this should not impact comparative results, it would have been more satisfactory to use baseline architectures that match the state-of-the-art.

Additional experiments or discussions on the following would greatly help:
-	As the spline method is not always better than Temp scaling, how do they compare from a computational viewpoint? How long does the binary search over thousands of calibration examples take compared to the DNN feed-forward?
-	From Tables 1,2 and 6, KS error and ECE rank methods quite differently. One reason one should trust KS more is that it does not depend on binning choices, and rely on a time-proven test. But could one come up with an experiment that shows that KS is provably more reliable?

Post rebuttal: I have read the authors responses with to my 4 questions, and appreciate how detailed and honest they are. They satisfy my concerns.

---

> ### Author Response · Authors · 2020-11-20
> **AR2: Thank you for the positive feedback.**
>
> We appreciate that the reviewer finds our method to be original, technically correct, and well written. Below, we address the reviewer’s questions/comments on some experiments and we have included these discussions in the updated version of the paper. If any answer needs further clarification to the reviewer, we'll make the best effort to improve the clarity.
>
> ### Our results vs Kull et. al.
> - We would like to clarify that our results _do not contradict_ with the results reported in Kull et al. As noted by the reviewer, Kull et. al. only report the average classwise ECE (cw-ECE) in the main paper (Table 3) whereas we report top-1 calibration error. The top-1 ECE results are reported in Table 15 in Kull et. al., which clearly demonstrates that ODIR methods are inferior to temperature scaling in this measure as observed by our experiments. We have made this clear in the updated manuscript, by explicitly referring to Table 15.
>
> ### Calibration on state-of-the-art networks
> - As noted by the reviewer, the trained models are obtained from Kull et. al. to be consistent with the reported numbers. Nevertheless, as requested by the reviewer, we compared our method against temperature scaling on a state-of-the-art network i.e. WideResNet-28-10 trained on CIFAR-100 (achieving 80.1% accuracy) and the results are as follows:
> |Calibration Metric|Uncalibrated|Temp. Scaling|Ours(Spline)
> |-|-|-|-|
> |KS (top-1)|0.058367|0.023443|**0.015145**|
> |ECE (25 bins)|0.062159|0.034364|**0.018672**|
>
> ### Running time of the spline method
> - The running time of our spline method (also temperature scaling) is negligible compared to one forward pass through the network. The most complex step of our method is sorting which is indeed very efficient. For example, on the calibration set of ImageNet with 25000 images, spline fitting takes only about 0.4 seconds. Note, both temperature scaling and our approach are post-hoc calibration methods that can be applied offline (very efficiently) once logits of the network are stored for calibration set.
>
> ### KS vs ECE
> - As discussed in the paper and noted by the reviewer, binning in ECE has multiple issues such as dependency on the number of bins, non-equal number of samples in each bin, etc. Clearly, being independent on the binning scheme, KS is more reliable. We believe designing an experiment to show the reliability of KS is an interesting open research problem and we are unable to provide a concrete answer as of now.

---

### Public Comment · ~Jize_Zhang1 · 2020-11-16
**Sensitivity of KS test & one related work**

Really like the idea on using KS test on calibration! My only conservation about KS is about its sensitivity, i.e., the KS discrepancy can be high if the considered distributions are both highly peaked, even if they are otherwise very similar. I'm wondering if this could be an issue in the calibration context?

I also want to refer to our recent work on an alternative binning-free ECE estimator [1]. As you have discussed in the intro, there exist several issues of histogram-based ECE estimators. In our work, we replaced histogram with KDE and provided a more reliable evaluation of calibration while mitigating the bias & binning sensitivity of histograms. The code is also available online if you are interested.

[1] Jize Zhang, Bhavya Kailkhura, and T Han. "Mix-n-Match: Ensemble and compositional methods for uncertainty calibration in deep learning.", ICML 2020, https://arxiv.org/pdf/2003.07329.pdf

---

> ### Author Response · Authors · 2020-11-24
> **Thanks for your interest in our paper.**
>
> We appreciate that the reader liked our idea of KS based calibration measure. Below we answer the reader's questions.
>
> ### Comparisons using KDE-ECE
> - Thanks for pointing us to your paper. We evaluated different calibration methods for networks trained on ImageNet dataset using KDE-ECE. We have added these results in Table 7 in our paper. Under evaluations with KDE-ECE, our method outperforms baseline methods.
>
> ### Sensitivity of KS
> - Since KS is the maximum difference between two CDFs, it is quite a robust estimate of calibration error. Infact irrespective of whether the distribution is peaky, KS is the same as the expected calibration error when one CDF consistently lies above the other (that is usually the case), which is what we want to measure. We would welcome any thoughts on this.

---

### Public Comment · ~Tiago_Salvador1 · 2020-11-17
**Questions from an interested reader**

It was quite enjoyable to read the paper. When reading it, I had the following two questions:
- The results presented for Imagenet use 25000 images for the calibration set which could potentially be one of the disadvantages of the method. How do the results change with fewer images, say only 5000 images?
- In temperature scaling, the parameter T is chosen to minimize the NLL on a calibration set. Could one instead minimize the KS error?

---

> ### Author Response · Authors · 2020-11-23
> **Thanks for your interest in our paper.**
>
> We appreciate that the reader found our paper interesting and enjoyable to read. Below we answer the reader's questions.
>
> ### Spline fitting with less samples on ImageNet
> - The only requirement for spline fitting to work well is to have smooth CDF. To achieve this, even a smaller number of samples would be sufficient.
> - We ran few experiments for DenseNet-16 trained on ImageNet to see how our calibration method works if we randomly chose only a subset of samples for spline fitting instead of 25000 samples in calibration set and the results are reported below:
> |# of samples|KS|
> |-|-|
> |25000|0.00406|
> |15000|0.00421|
> |5000|0.00448|
> - It can be clearly observed that our method is quite robust in terms of number of samples used in the calibration set consistently beating all the baselines even using only 5000 samples in the calibration set. The second best method in this case is temperature scaling which achieves KS error of 0.00744 with 25000 samples in the calibration set.
>
> ### Temperature scaling on other metric instead of NLL
> - In this paper, we have followed Kull et al. implementation of temperature scaling which minimizes NLL but one can choose to minimize calibration metrics such as ECE or KS.
> - We believe that the main limitation of using temperature scaling is that it can only estimate the recalibration function using a single parameter which may not be sufficient in many cases.

---

### Decision · Program_Chairs · 2021-01-07
**Final Decision**

**Decision:**

Accept (Poster)

**Comment:**

Many papers have been written on calibrating neural networks recently.  This paper presents a definition of calibration that is more robust than the popular ECE measure while also being more discerning than the Brier score.  Then it proposes a practical spline-based method of post-editing the output softmax scores to make them more calibrated.  The method is shown to be better than existing methods both on their measure and established measure (thanks to reviewer's questions on that.).
The paper should be of much interest to the community.